# Barriers to effective hypertension management in rural Bihar, India: A cross-sectional, linked supply- and demand-side study

**Michael A. Peters** [1]*, **Olakunle Alonge** [1], **Anbrasi Edward** [1], **Yvonne Commodore-Mensah** [2], **Japneet Kaur** [1], **Navneet Kumar** [3], **Krishna D. Rao** [1]

**1** Department of International Health, Johns Hopkins Bloomberg School of Public Health, Baltimore, MD, United States of America, **2** Johns Hopkins School of Nursing, Baltimore, MD, United States of America, **3** Oxford Policy Management, New Delhi, India

* mpeters@jhu.edu

## Abstract

Effective management of hypertension in low- and middle-income settings is a persistent public health challenge. This study examined supply- and demand-side barriers to receiving quality care and achieving effective hypertension management in rural Bihar, India. A state-representative household survey collected information from adults over 30 years of age on characteristics of the hypertension screening, diagnosis, and management services they received. A linked provider assessment determined the percent of providers who provided quality hypertension care (i.e., had a functioning BP measurement device, measured a patient's BP, could correctly diagnose hypertension, had at least one first-line antihypertension medication, and could prescribe correctly based on standard guidelines). Patients were linked with their provider to determine the quality-adjusted coverage of hypertension management and logistic regression analysis was conducted to determine characteristics associated with receiving quality care. A total of 14,386 patients and 390 providers were studied. Nearly a quarter (22.5%) of adults had never had their BP measured before and 8.1% of adults reported a previous hypertension diagnosis. Less than one third (31.0%) of all interviewed providers demonstrated ability to provide quality hypertension care, and quality varied between provider types (14.8% of private homeopathic, 25.2% of informal, 40.0% of private modern medicine, and 60.0% of public providers gave quality care). While 95.8% of diagnosed individuals received some treatment, only 10.9% of patients received care from quality local providers. Nearly 45% of individuals with hypertension received care from non-local providers. Individuals from the general caste with comorbidities living in villages with more high-quality providers were most likely to receive quality care from a local provider. Whereas the coverage of services for individuals diagnosed with hypertension is high, the quality of these services is suboptimal for economically and socially vulnerable populations, which limits effective management and control of hypertension in rural Bihar. Efforts should be targeted towards providers to initiate quality treatment upon diagnosis, including correct prescription of antihypertensives.

**Data Availability Statement:** The data used in this study are owned by a research consortium, led by the Government of Bihar. Permission has not been

given by the government to deposit the raw data in a public repository. The research consortium (and government) has reviewed and approved the publication of the results included in this study. The research consortium that sponsored this study was chaired by the Government of Bihar, and included Johns Hopkins, the Bill and Melinda Gates Foundation, CARE India, and Oxford Policy Management. Data collected through the study is the property of the Government of Bihar. Please contact the corresponding author with requests to access the data, which will be elevated to the Government of Bihar for consideration, or direct data requests to the contact person for the Preva Group who runs the Bihar Collaborative, Sahil Mahajan (smahajan@prevagroup.com).

**Funding:** KDR received funding through the Bill and Melinda Gates Foundation to conduct this research under grant #OPP119434. The funder played no role in the design of the study nor the collection, analysis, and interpretation of data nor the writing of the manuscript.

**Competing interests:** The authors declare that they have no competing interests.

## Introduction

Non-communicable diseases (NCDs) are responsible for two-thirds of all mortality in low- and middle-income countries (LMICs), including nearly 18 million deaths from cardiovascular disease (CVD) annually [1]. Hypertension is the leading CVD risk factor, and improving hypertension management in primary care is key for achieving universal health coverage [1–4]. With the rising burden of hypertension and other NCDs, health systems in LMICs are challenged to adapt care models to address these lifelong conditions, while also maintaining services to reduce the impact of infectious diseases and maternal and childhood conditions [5]. To date, health systems in LMICs have largely been unable to provide health services at sufficient coverage or quality to prevent deaths from NCDs [6]. Each year, 2.4 million preventable CVD deaths are caused by poor-quality health services, more than five times the CVD deaths attributed to the lack of access to health services [6]. Research is needed to inform ways of improving the quality of health services in LMICs, especially services for preventing and managing NCDs, such as hypertension-related CVD. Efforts to identify gaps in the provision of quality hypertension management along the care continuum (from prevention, to screening and diagnosis, to management) can inform the design of health systems and programs that better respond to NCDs.

One way of assessing disease-specific health system functioning is to separate performance in a stepwise fashion along the continuum of care into a care cascade. The traditional hypertension cascade of care has steps for the prevalence of hypertension, the percent of those people who are aware of their diagnosis, the percent who are receiving treatment, and finally, those who have controlled blood pressure (BP). This hypertension care cascade framework has been used to describe BP control for decades and demonstrates that the largest barriers to hypertension management in LMICs are the diagnosis of individuals with elevated BP, and the ability to achieve non-elevated BP among those who have initiated treatment [7–10]. Most research on the hypertension care cascade relies exclusively on household surveys and population estimates, and does not provide insight into the quality of care received, therefore quality-adjusted coverage (or effective coverage) of hypertension services is largely unknown.

Without adjusting for a measure of quality, service coverage is only weakly linked to the health benefits experienced by a population [11]. There are several methods for calculating quality-adjusted coverage, including linking the content of care with population estimates of coverage, yet few studies have applied this approach to studying hypertension management [12]. A recent literature review found 12 studies which examined processes of quality care in population-based studies of hypertension management and only 8 studies which attempted to describe effective coverage of hypertension management in LMICs [13]. Estimates of quality-adjusted coverage of hypertension management services have immediate policy relevance for contexts experiencing epidemiological transition, like Bihar, India, which must prepare for an increasing burden of hypertension.

Bihar is facing a double burden of persistently high prevalence of infectious diseases and rapidly increasing incidence of chronic illnesses [14]. Hypertension is the leading CVD risk factor for death in the state, and was responsible for 6.4% of disability-adjusted life years lost in 2016, more than a two-fold increase from 2.8% in 1990 [14]. While studies have examined the quality of care provided by India's pluralistic health systems, little is known about how effective public and private primary care providers are in the context of diagnosing and treating hypertension [15, 16]. The National Programme for prevention and control of cancer, diabetes, cardiovascular diseases and stroke (NPCDCS) is the main program to address NCDs across India, and a recent review highlighted the need to strengthen public health facilities to provide screening, early diagnosis, and treatment [17]. In Bihar, the coverage of screening and

treatment services is low: less than half (46.8%) of hypertensive individuals were recently screened and aware of their high blood pressure status, and only 17.3% were receiving treatment [18]. On the other hand, the quality of hypertension management services in Bihar is largely unknown, especially in rural areas where informal providers (IPs) are more prevalent. This is significant because perceived quality of care is a major driver of provider choice, or switch, among Indian adults with hypertension [19]. This often leads to patients bypassing local care options to receive services that may be far away, but are more acceptable [20]. Understanding who is bypassing care is essential for making health systems more people-centered, equitable, and efficient.

Quantifying supply- and demand-side factors that prevent effective hypertension management can inform future policy for improving BP control in primary health care settings in Bihar and other LMICs. This study describes population-level (demand-side) deficits along the care cascade in Bihar, and (supply-side) gaps in the quality of hypertension care across provider types in India's pluralistic health system. It estimates the quality-adjusted coverage of hypertension management services in Bihar and determines factors associated with receiving high quality care from local providers and with bypassing local care options. Taken together, this study helps to advance the development of high-quality health systems for addressing hypertension and NCDs in LMICs.

## Materials and methods

### Study context

With a population of 104 million people at the 2011 Census, Bihar is India's third most populous state [21]. The health system is characterized by unequal access to health care, insufficient human resources and institutional capacity, poor quality care, and high out-of-pocket health expenditure: the state has the second highest ratio of private to public spending on health care in India [22]. This is partly caused by Bihar's pluralistic primary health care system, in which nearly 75% of care is provided by the private sector. The private sector includes (i) providers practicing modern medicine with a Bachelor of Medicine, Bachelor of Surgery, or MBBS degree, (ii) providers trained in Indian systems of medicine with a degree in Ayurveda (Bachelor of Ayurvedic Medicine and Surgery), Yoga and Naturopathy (Bachelor of Yoga), Unani (Bachelor of Unani Medicine and Surgery), Siddha (Bachelor of Siddha Medicine and Surgery), and Homeopathy (Bachelor of Homeopathic Medicine and Surgery), collectively known as AYUSH providers, and (iii) informally trained providers (IPs) [23]. IPs are often trusted members from the community in which they serve, with limited health training or experience, charge nominal fees (generally 100 Rupees, or $1.37 USD per consultation), and are an important source of primary care services in rural Bihar and other parts of India, though their role has evolved differently in various market settings [24]. A recent nationwide study showed that in Bihar, 3.9% of villages had access to a public MBBS provider, 7.6% had access to any MBBS provider (public or private), and 96.2% of villages had access to any provider, including AYUSH or informal providers [16].

The public sector infrastructure includes about 1,800 Primary Health Centers (PHC), which represents about 60% of the PHCs required to serve Bihar's population based national government estimates for infrastructure allocation [22]. PHCs are staffed by MBBS or AYUSH doctors, but weaknesses in the public sector caused by staff shortages and a lack of basic infrastructure and supplies further reinforce private health care-seeking behavior. Among people who seek care for chronic conditions in Bihar, 86% receive care from the private sector (39% from private MBBS and AYUSH providers, 30% from IPs, and 17% from pharmacists and other private sources), leaving only 14% who receive care from the public sector [25].

## Study design

This cross-sectional study used primary data collected from a household survey and a provider quality assessment under the Assessment of Bihar's Primary Health Care System (ABPHC) study. The study's sample size was designed to detect a difference in the proportion of patients who visit PHCs with and without competent clinicians, and required a total of 9,798 households, accounting for design effect, the prevalence of illness and care-seeking, and average household size (S1 Annex). The ABPHC household survey employed a multi-stage sampling design, with rural PHCs as the primary sampling unit, villages within the PHC catchment area as the secondary sampling unit, and households within the village as the tertiary sampling unit. Rural PHCs were randomly selected from a census using stratified sampling proportional to the number of PHCs in Bihar's nine divisions. Villages were randomly sampled by probability proportional to population size from a census of villages within each selected PHC's catchment area, and 30 households were randomly sampled from each selected village using a complete listing of households. The total envisioned sample size was 10,500 households. All consenting members of selected households were included in the study and were administered a standard questionnaire in the local language by trained enumerators. The questionnaire included sections on demographic information, illness, and care-seeking history and experiences. Individuals aged 30 and older were asked about care-seeking related to specific chronic diseases (hypertension, diabetes, chronic heart disease, asthma, and chronic obstructive pulmonary disease). Data was collected between November 2019 and March 2020. Responses from the household survey informed sampling for the provider assessments.

Local public and private care providers (including MBBS, AYUSH, and IPs) visited by household survey respondents from 70 randomly selected villages (1 from each PHC catchment area) were located and included in the supply-side provider quality assessment if they were within five kilometers of the village. The study sample size was calculated to detect differences in the quality of public and private providers, and required 67 providers in each group (S1 Annex). The provider assessment was administered by enumerators with nursing degrees and consisted of three parts: a facility readiness assessment, responses to four clinical vignettes, and direct patient observations. The facility readiness assessment was modified from the validated Demographic Health Survey Service Provision Assessment survey, and was designed to understand the medicines, equipment, and human resources available to providers as compared to Bihar's Essential Medicines List [26, 27]. In the clinical vignettes, hypothetical patients and scenarios were described to providers to assess knowledge on how to treat certain conditions (hypertension diagnosis, child diarrhea, child pneumonia, and angina). Providers were prompted to ask questions about the patient's history, list the tests they would conduct on the patient, make a diagnosis, describe the advice they would give the patient, and if necessary, write a prescription for the patient. To understand provider practice, nurse enumerators observed patient-provider interactions and recorded provider actions using a standardized form. Enumerators stayed with the provider for three hours, observed up to five new patient consultations, and collected information on the same categories used in the clinical vignettes (i.e., patient history, tests conducted, diagnosis, advice given, and prescription). Data collection for the provider assessment started in January 2020, was interrupted in March due to the SARS-CoV-2 (COVID-19) outbreak, and then completed between February and March 2021. Due to the continuing spread of COVID-19, when data collection resumed in 2021, the patient observation component of the provider assessment was eliminated from the study to ensure safety of enumerators and patients.

All tools were pilot tested in non-sampled villages in rural Bihar to ensure that questions captured intended constructs. Wherever possible, the household and provider assessment

tools used validated questions from surveys implemented in the same context, including the National Family Health Survey, the National Sample Survey, and World Bank Health Care Provider Surveys [15, 28, 29]. The hypertension vignette was designed in consultation with cardiologists and pilot tested with local primary care providers in Bihar for psychometric validation. Data for the household survey and provider assessment was collected on tablets using SurveySolutions, a free Computer Assisted Personal Interviewing (CAPI) software to reduce information bias [30]. Enumerators had previous experience in household survey data collection and participated in a week-long training to familiarize themselves with the tool, the CAPI platform, and to participate in large scale field tests. Direct supervision and random response reliability checks were conducted in the field to ensure data quality and reduce the risk of measurement bias. All data was passed through automatic logic checks and implausible responses were discussed and resolved between supervisors and enumerators.

Data from the household survey and the provider assessment was linked through a direct matching process. Adults with hypertension were matched with the providers from whom they received care. Individuals were not linked to their providers if they (i) sought care more than five kilometers away from their village (because providers were not eligible for the provider assessment), (ii) sought care from medical shops (because these providers were unable to complete the provider assessment during pretesting), or (iii) sought care from a provider within five kilometers of the village but the provider was not located by the study team.

The ABPHC study is approved for human subjects research by the Johns Hopkins University Institutional Review Board (IRB00009563) and by the Sigma Institutional Review Board in India (Doc#'s 1910787106 / 1910789424). Written consent to participate was obtained from all study participants.

## Study measures

Since the ABPHC study did not measure respondent BP, a traditional care cascade with a common denominator of all people with high BP could not be created [31]. Instead, population-based estimates of hypertension care cascade steps were calculated using a previously defined methodology [18]. Respondents were considered to have been "screened" if they responded "yes" to the question, "Have you ever had your blood pressure measured by a doctor or other health worker?". Respondents were "aware" if they responded "yes" to the question, "have you been told by a doctor or other health worker that you have high blood pressure or hypertension?". Participants were "treated" if they responded "yes" to the question, "Have you taken any medications or other treatment for high blood pressure during the last 12 months?". Additional supply-side measures included the source of care for hypertension management, reasons for choosing a provider, self-reported distance traveled to seek hypertension care, monthly amount spent on hypertension care, annual number of visits to a provider for hypertension management, and hypertension-related hospitalizations.

Measures of provider quality were described chronologically in terms of their impact on a patient's ability to progress through the hypertension care cascade, from screened to aware to treated and controlled BP. The definition of controlled blood pressure varies based on context and clinical guidelines. For the purposes of this study, hypertension is defined as systolic blood pressure of 140 mm Hg or more and/or diastolic blood pressure of 90 mm Hg or more and/or taking antihypertensive medication after a previous diagnosis, in accordance with International Society of Hypertension guidelines and the Fourth Indian Guidelines on Hypertension. For example, a provider's ability to appropriately diagnose hypertension based on a high blood pressure reading contributes to a patient's ability to progress from "screened" to "aware". A cascade of provider ability to manage a patient with hypertension was described based on

measures from hypertension vignettes and facility assessments (S2 Annex). Providers who made it to the final cascade step (had a functioning BP measurement device, would check a hypothetical patient's BP at least once, made a correct hypertension diagnosis, initiated medical treatment upon diagnosis of stage two hypertension, had first line anti-hypertensive medication available, and wrote an appropriate prescription) are considered to deliver high quality hypertension care. The appropriateness of prescriptions was assessed by three clinicians who have practiced primary care medicine in India and other middle-income settings. Two clinicians reviewed each prescription and rated their quality on a scale including (i) inappropriate and harmful, (ii) inappropriate but not harmful, and (iii) appropriate for safely lowering BP using a standardized protocol (S3 Annex). Disagreements in ratings were discussed as a team and resolved by the third clinician. Additionally, the percent of adults over 30 who were screened by a health worker in patient observations is presented to demonstrate provider actions in practice.

In the linked supply- and demand-side analysis, we described the effective coverage of hypertension management services, defined by the proportion of hypertension-aware respondents who received high-quality care. We examined how the probability of receiving quality care and bypassing varied by the following characteristics: age, sex, caste, wealth index, education, the presence of a comorbidity, and the number of high-quality providers of hypertension care in a village. Hypertension-aware patients were considered to have bypassed local options if they reported traveling more than one kilometer farther than the local PHC to receive hypertension care. Age was grouped into 10-year categories starting at age 30, until all individuals over age 70. Caste was grouped by self-reported membership of head of household according to the Government of India's categorization scheme: scheduled castes (SC), scheduled tribes (ST), other backward castes (OBC) and general. Household wealth quintile was calculated using principal components analysis on ten household assets following a standard methodology [32]. Education was categorized into three groups: individuals with no schooling completed, individuals with some schooling completed but less than secondary high schooling, and secondary high schooling completed and above. Individuals were determined to have a comorbidity if they responded that they had been told by a health worker that they had diabetes, asthma, chronic heart disease, or cancer in addition to hypertension.

## Statistical analysis

Normalized weights were calculated for estimates from the household survey to account for probability of selection through the ABPHC study's design and to account for non-response rates. All descriptive statistics for population estimates from the ABPHC household survey incorporated these weights, while estimates from the provider surveys were unweighted due to the lack of a complete sampling frame of private providers. To account for the variability in private provider qualifications, supply-side results from the private sector are presented disaggregated by provider training (i.e., MBBS, AYUSH, and IP).

Logistic regression analyses were employed to determine predictors of the probability of receiving quality care in the village and the probability of bypassing local care options. Individuals with missing data are excluded from analysis and sample sizes for each regression are discussed further in S4 Annex. For each regression analysis, collinearity of explanatory variables was assessed, a likelihood ratio test was conducted for each variable, effect modification was considered and tested, and the final model was compared with stepwise model selections. Hosmer-Lemeshow tests for goodness of fit were conducted and regression diagnostics were conducted to assess final model fit. Data analysis were performed in StataSE version 14.1 (College Station Texas) [33].

## Results

### Sample characteristics

In total, 39,486 individuals were included in the household survey, including 14,386 individuals who were 30 years or older and answered the chronic disease module and 950 individuals who reported a previous hypertension diagnosis (Table 1).

The provider assessment included 390 providers from 70 villages, 368 of whom (94.4%) said they would treat the patient in the hypertension vignette (Table 2). In total, our study administered the provider assessment to 83.0% of the local providers who provided care to individuals in the household survey. This compares favorably with a previous study in India with a similar study design, which reported a response rate of 41.9% among local providers [16].

### Population measures of hypertension management

Of the 14,386 individuals in the survey aged 30 or older, 77.5% had ever had their BP measured by a health worker before the survey date (95% CI: 76.3% to 78.6%). Among adults who had

**Table 1. Characteristics of household survey respondents.**

|  | All respondents (N = 39,486) | Respondents over 30 (N = 14,386) | Respondents aware of hypertension diagnosis (N = 950) |
|---|---|---|---|
| **Sex** | | | |
| Female | 20,767 (52.6%) | 7,742 (53.8%) | 551 (58.0%) |
| Male | 18,719 (47.4%) | 6,644 (46.1%) | 339 (42.0%) |
| *Missing* | 0 (0.0%) | 0 (0.0%) | 0 (0.0%) |
| **Age** | | | |
| Average (SD) | 25.8 (20.2) | 48.6 (13.9) | 58.0 (13.4) |
| **Religion** | | | |
| Hindu | 35,541 (90.0%) | 13,066 (90.8%) | 839 (88.3%) |
| Muslim | 3,909 (9.9%) | 1,307 (9.1%) | 110 (11.6%) |
| Other (Christian, Buddhist Jain) | 26 (0.1%) | 8 (0.1%) | 0 (0.0%) |
| *Missing* | 10 (<0.1%) | 5 (<0.1%) | 1 (<0.1%) |
| **Caste** | | | |
| Scheduled Caste | 10,252 (26.0%) | 3,453 (24.0%) | 139 (14.7%) |
| Scheduled Tribe | 544 (1.4%) | 180 (1.3%) | 10 (1.0%) |
| Other Backward Caste | 24,901 (63.1%) | 9,094 (63.2%) | 596 (62.8%) |
| General | 3,779 (9.6%) | 1.654 (11.5%) | 204 (21.5%) |
| *Missing* | 12 (<0.1%) | 12 (<0.1%) | 0 (0.0%) |
| **Highest level of Education Achieved** | | | |
| No schooling | 13,115 (33.2%) | 8,072 (56.1%) | 504 (53.1%%) |
| Some schooling | 18,708 (47.4%) | 5,034 (35.0%) | 349 (36.7%) |
| Higher secondary schooling or above | 4,039 (10.2%) | 1,279 (8.9%) | 97 (10.2%) |
| *Missing*+ | 3,624 (9.2%) | 1 (<0.1%) | 0 (0.0%) |
| **Occupation** | | | |
| Not employed | 30,225 (76.6%) | 7,350 (51.1%) | 609 (64.1%) |
| Agriculture | 3,111 (7.9%) | 2,727 (18.9%) | 176 (18.5%) |
| Labor | 3,346 (8.5%) | 2,282 (15.9%) | 45 (4.7%) |
| Self-employed | 1,920 (4.9%) | 1,453 (10.1%) | 81 (8.5%) |
| Service/salaried | 875 (2.2%) | 572 (4.0%) | 39 (4.1%) |
| *Missing* | 9 (<0.1%) | 2 (<0.1%) | 0 (0.0%) |

+Question not asked to children under the age of 5; SD = Standard Deviation

**Table 2. Characteristics of providers (N = 390).**

|  | Private Providers (N = 319) | Public Providers (N = 71) |
|---|---|---|
| **Sex of Provider** | | |
| Male | 311 (97.5%) | 62 (87.3%) |
| Female | 7 (2.2%) | 9 (12.7%) |
| *Missing* | 1 (0.3%) | 0 (0.0%) |
| **Age of Provider** | | |
| Average (SD) | 45 (14) | 48 (11) |
| **Caste of Provider** | | |
| Scheduled Caste | 31 (9.7%) | 7 (9.9%) |
| Scheduled Tribe | 11 (3.5%) | 4 (5.6%) |
| Other Backwards Caste | 165 (51.7%) | 31 (43.7%) |
| General | 110 (34.5%) | 28 (39.4%) |
| Other | 2 (0.6%) | 1 (1.4%) |
| **Place of Residence** | | |
| Same Town/Village as clinic | 238 (74.6%) | 47 (66.2%) |
| Other town/village | 81 (25.4%) | 24 (33.8%) |
| **Provider Training** | | |
| MBBS | 15 (4.7%) | 47 (66.2%) |
| AYUSH | 54 (16.9%) | 18 (25.4%) |
| Informal training | 250 (78.4%) | 0 (0.0%) |
| Other modern medicine* | 0 (0.0%) | 6 (8.5%) |
| **Years worked at facility** | | |
| Average (SD) | 15 (13) | 6 (6) |

SD = Standard Deviation;

*: Other modern medicine trainings include degrees in dentistry and physiotherapy

ever had their blood pressure measured before, 8.1% reported a previous hypertension diagnosis. The prevalence of a previous hypertension diagnosis increased with age with the highest prevalence (15.2%) among those 70 and older. The majority (95.7%) of the hypertension-aware population was treated. Most treated individuals (94.4%) availed services from the private sector (46.4% from private doctors and clinics, 18.9% from pharmacies or compounders, 14.5% from private hospitals, 14.3% from traditional healers, and 0.3% from other private providers). Among the 5.6% of individuals who sought care from the public sector, the majority (58.8%) sought care from district hospitals. The perceived quality of hypertension management services was an important consideration for individuals in their care-seeking decisions. Of the individuals who sought hypertension management services from the private sector, 62.2% did not seek hypertension treatment from their local PHC due to the low perceived quality of care. This contributed to people traveling an average of 29.2 kilometers (95% CI: 20.9 to 37.5 kilometers) to seek hypertension management services. On average, people who sought care outside of the home to manage their hypertension made 4.8 visits per year to a provider. The median monthly amount spent on hypertension management was 250 rupees (US $3.35). The median monthly spending is slightly higher for those who receive treatment from private providers than public providers (255 vs 200 rupees). Among respondents who were aware of their high BP, 9.5% had ever been hospitalized due to their hypertension. This includes 4.6% of patients with hypertension who were hospitalized in the past year for chest pain, cardiovascular problems, or hypertension.

## Provider measures of hypertension management

Among the 390 sampled primary care providers, 368 (94.4%) treated hypertension, and 121 (39.0%) could provide high-quality hypertension management. The largest system-wide gaps in the provision of high-quality hypertension management were the ability of providers to make a correct hypertension diagnosis, the availability of at least one first line antihypertensive medication, and the ability to write an appropriate prescription to safely lower BP (Fig 1). Across provider types, public providers were most frequently able to provide quality hypertension care (62%), followed by private MBBS-trained providers (40%), IPs (25%), and private AYUSH providers (15%) (Table 3). The largest gap was the lack of any first-line antihypertensive medication at AYUSH provider-run clinics. Among IPs, the largest single barrier to providing quality care was the ability to make a correct hypertension diagnosis. For both public and private MBBS providers, the largest gap in quality care was the ability to write an appropriate prescription to manage hypertension (drop of 15% and 20%, respectively). Among providers who correctly diagnosed the hypothetical patient as hypertensive, 16.6% of private MBBS doctors, 13.2% of private AYUSH doctors, 10.6% of IPs and, and 9.7% of public providers wrote prescriptions that were assessed to cause harm to the patient. Prescriptions were characterized as harmful because they included inappropriate drugs (e.g., Ampicillin, Betnisole, or Paracetemol), excessive doses, and/or harmful drug interactions.

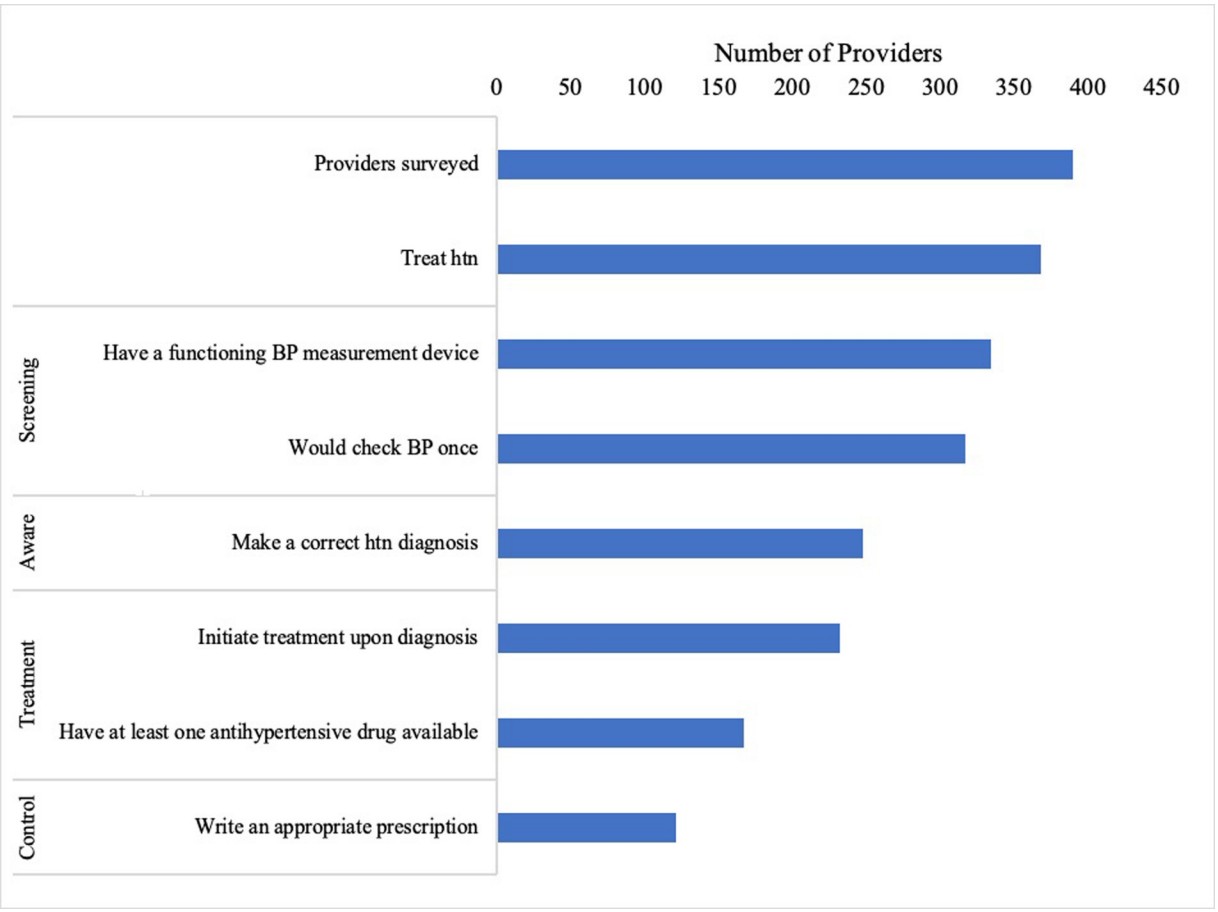

**Fig 1. Provider cascade of quality hypertension care.** BP = blood pressure; HTN = hypertension.

**Table 3. Provider cascade of quality hypertension care by provider type.**

| | Public (N = 71) | Private - MBBS (N = 15) | Private - AYUSH (N = 54) | Private - IP (N = 250) | Total (N = 390) |
|---|---|---|---|---|---|
| Providers surveyed | 100.0% | 100.0% | 100.0% | 100.0% | 100.0% |
| Treat HTN | 100.0% | 93.3% | 98.1% | 92.0% | 94.4% |
| Have a functioning BP measurement device | 98.6% | 93.3% | 85.2% | 81.6% | 85.6% |
| Would check BP once | 95.8% | 93.3% | 81.5% | 76.4% | 81.3% |
| Make a correct HTN diagnosis | 83.1% | 80.0% | 63.0% | 57.2% | 63.6% |
| Initiate treatment upon diagnosis | 81.7% | 80.0% | 63.0% | 51.2% | 59.5% |
| Have at least one antihypertensive drug available | 77.5% | 60.0% | 18.5% | 37.2% | 42.8% |
| Write an appropriate prescription | 62.0% | 40.0% | 14.8% | 25.2% | 31.0% |

BP = blood pressure; HTN = hypertension;

Across patient observations, 30.1% of patients over the age of 30 had their BP measured during a consultation. Patients presenting to private primary care providers were significantly more likely to have their BP measured than those presenting to public providers (36.5% versus 19.7%, p = 0.02). Among private providers, formally trained providers (MBBS providers checked BP in 70.0% of visits, AYUSH providers checked BP in 42.9% of visits) checked adult BP more frequently than IPs (31.0% of visits).

### Linked assessment of hypertension management

In the 70 villages where the provider assessment was conducted, 192 individuals self-reported a previous hypertension diagnosis. Of these 192 individuals, 85 sought care from providers outside the village (more than 5 kilometers away), 14 sought care from medical shops, and 19 sought care from local providers who were not assessed. Local providers were not included in the assessment because (iii) the provider was out of town for an extended period (n = 9), (ii) the provider was not located from information provided by the individual (n = 7), and (iii) the individual did not remember the name of their provider (n = 3). In total, the quality of care received is known for 74 individuals who received care from 56 local providers (S4 Annex). Of these individuals, 83.8% sought care from IPs, 8.1% sought care from private AYUSH doctors, 4.0% sought care from private MBBS doctors, and 4.0% sought care from public providers. By caste, 93.3% of SC/ST individuals sought care from IPs compared with 80.0% of OBC and 85.7% of General caste individuals.

Over one-third of hypertension-aware patients (34.9%) received care from a local provider who did not provide quality care, including 7.3% who sought care from medical stores whose operators were unable to complete the provider assessment due to a lack of medical knowledge. Of the 192 adults with hypertension, 10.9% received quality care from local providers, and another 44.3% received care from non-local providers. The quality-adjusted coverage of hypertension management services in rural Bihar ranges from 10.9% to 65.1% if unlocated local and out of market providers all provide quality care.

Among the 73 individuals whose quality of care was known and were included in the regression analysis, having a comorbidity and the number of quality providers in a village were statistically significantly associated with a higher probability of receiving quality care from a local provider (Table 4). Additionally, people from the general caste were significantly more likely to receive quality care from a local provider than those from the SC caste. Among individuals in the villages where quality of care was known, the number of quality providers in a village was not associated with the probability of bypassing local care options. In other words, there was no association between the number of local high-quality providers and the

**Table 4. Determinants of receiving quality care from local providers and bypassing among hypertensive adults in Bihar.**

| Outcome variable | (1) Receiving local quality care | (2) Bypassing local care in villages with quality assessment | (3) Bypassing local care in all villages |
|---|---|---|---|
| **Sex** | | | |
| Female | Ref | Ref | Ref |
| Male | 0.168 | 0.682 | 0.987 |
| | [0.0195,1.447] | [0.313,1.484] | [0.712,1.368] |
| **Age Group** | | | |
| 30–39 | - | Ref | Ref |
| 40–49 | - | 0.189* | 0.883 |
| | | [0.0408,0.881] | [0.491,1.589] |
| 50–59 | - | 0.404 | 0.992 |
| | | [0.0926,1.767] | [0.561,1.757] |
| 60–69 | - | 0.502 | 1.289 |
| | | [0.120,2.100] | [0.750,2.214] |
| 70 and older | - | 0.715 | 1.238 |
| | | [0.164,3.123] | [0.695,2.203] |
| **Caste** | | | |
| SC | Ref | Ref | Ref |
| ST | NA (omitted) | 6.354 | 1.925 |
| | | [0.675,59.78] | [0.494,7.504] |
| OBC | 39.88 | 3.258* | 1.581* |
| | [0.840,1893.3] | [1.080,9.833] | [1.031,2.424] |
| General | 83.05* | 3.292 | 1.972** |
| | [1.077,6407.1] | [0.900,12.04] | [1.186,3.279] |
| **Wealth Category** | | | |
| Q1 (Poorest) | Ref | Ref | Ref |
| Q2 | 21.27 | 2.223 | 1.638* |
| | [0.761,594.7] | [0.673,7.343] | [1.028,2.610] |
| Q3 | 1.711 | 0.765 | 1.102 |
| | [0.157,18.69] | [0.261,2.240] | [0.694,1.750] |
| Q4 | 2.679 | 1.193 | 1.180 |
| | [0.137,52.24] | [0.388,3.672] | [0.729,1.911] |
| Q5 (Richest) | 1.855 | 1.157 | 1.860** |
| | [0.152,22.60] | [0.416,3.218] | [1.187,2.915] |
| **Education** | | | |
| No schooling | Ref | Ref | Ref |
| Some schooling | 0.919 | 2.572* | 1.393 |
| | [0.132,6.403] | [1.111,5.954] | [0.992,1.957] |
| Higher secondary schooling or above | 1 | 10.83** | 1.941* |
| | [1,1] | [1.935,60.59] | [1.129,3.338] |
| **Comorbidity** | 16.08* | 2.000 | 2.358*** |
| | [1.249,207.2] | [0.953,4.197] | [1.722,3.231] |
| **Number of quality hypertension providers in village** | 8.128** | 0.933 | |
| | [2.121,31.15] | [0.664,1.312] | |
| N | 73+ | 192 | 949 |

Values presented are exponentiated coefficients of odds ratios; 95% confidence intervals in brackets; due to small sample size, interpretations of age groups are not presented for regression 1, probability of receiving local quality care.

* $p < 0.05$,

** $p < 0.01$,

*** $p < 0.001$

+ Of the 74 individuals whose quality of care was known, one individual from the ST caste reported being hypertensive, therefore this observation was dropped from the regression analysis to avoid an exposure that perfectly predicted the outcome. Final sample size for this regression is 73 individuals.

probability of bypassing local care. Among all hypertensive individuals, those with a comorbidity were most likely to bypass local care options. Additionally, individuals in the OBC or general caste, wealthier individuals, and people with higher secondary schooling or above were significantly more likely to bypass local care options than those in the ST caste, the poorest wealth quintile, and with no education, respectively.

## Discussion

This study describes the landscape of hypertension management in Bihar, India using linked supply- and demand-side estimates of quality care and receipt of care. Our findings reinforce previous population-based surveys by suggesting that most individuals in India with high BP have either never been screened or are not aware of their condition, (i.e., they have not been diagnosed as hypertensive) (S5 Annex) [18, 28]. Nearly one-quarter of adults in rural Bihar have never had their BP measured and just over eight percent of the adult population reported a previous hypertension diagnosis, compared to previous studies which reported a population prevalence of 12.8% in Bihar [18]. This screening gap within the health system is confirmed by patient observations, during which fewer than a third of adult patients had their BP measured by a provider. According to NPCDCS and other screening guidelines, all adults over the age of 30 are supposed to have their BP measured opportunistically and during each interaction with the public health system [34, 35]. Across provider types, the ability to make a correct hypertension diagnosis was the largest gap in the ability to provide quality hypertension management, which also contributes to the "awareness" gap in the population. Increasing the percent of individuals who are screened and aware is critical but is not sufficient to improve hypertension control in rural Bihar.

Most "aware" individuals received care, however the quality of this care was generally low. While 95.7% of adults who were aware of their condition received some treatment, the quality-adjusted coverage of hypertension management services by local providers was as low as 10.9%. Nearly half of individuals with reported hypertension bypassed local providers, traveling between 20 and 40 kilometers to receive care. Richer, more educated individuals from OBC and general castes and those with comorbidities were most likely to bypass local providers. Public providers were most frequently able to provide quality care across local provider types, yet most individuals perceived the quality of care at PHCs to be poor. Taken together, these results suggest that while access to hypertension management services is generally good, the quality of these services is inadequate in rural Bihar, and people who have the means to travel great distances to receive care prefer to do so rather than receive the more convenient care available in their villages. In short, local health systems are not providing high-quality, equitable services to manage hypertension in rural Bihar.

Findings from this study are subject to some important limitations. First, the ABPHC household survey design was cross-sectional and did not include direct measurements of BP. This means that all estimates among hypertensive individuals are based on self-reported hypertension diagnosis, likely resulting in underestimates of the true prevalence of hypertension. Previous studies have demonstrated validity between self-reported and true hypertension status, suggesting that this is an acceptable method for data collection [36]. Second, the clinical vignettes were not measurements of what providers did in practice and only five total consultations to manage hypertension were directly observed due to COVID-19 related interruptions to the study implementation. Information on the percent of providers that would measure BP twice during the vignette and give lifestyle advice to patients in the vignette were collected but were not included in the final definition of quality care because too few providers followed these measures. Studies have demonstrated correlations between provider competence and

provider actions–there is typically a gap between the two–however provider knowledge is generally regarded as the upper limit of what a provider can do, so our estimates of poor provider quality are likely to be conservative in nature, and the true quality is likely to be worse [37, 38]. Additionally, the study definition of delivering quality hypertension care may have influenced results, as AYUSH providers and IPs are not legally allowed to stock or prescribe medications. Given the importance of immediate antihypertensive therapy in improving health outcomes, we believe that the availability of antihypertensive medications is an appropriate indicator of structural quality of care [39]. Third, the study design was limited to assessments of providers utilized in local rural markets (within five kilometers of a sampled village). While most individuals with hypertension sought care from providers within five kilometers of their home (55.7%), many individuals also traveled long distances to receive treatment. The linked regression analysis had a small sample size resulting in large point estimates and wide confidence intervals; however, we were still able to understand characteristics of individuals who bypassed local care options for hypertension management.

The primary strength of this study is that it reveals new characteristics of hypertension management through a linked supply- and demand-study design which adds important information about the quality of hypertension care received by populations. This study adds insights about the health systems bottlenecks that prevent people with hypertension from achieving optimal health outcomes. Namely, poor structural quality (lack of front-line antihypertensive medications) and insufficient provider knowledge of how to properly diagnose and treat patients, combined with a lack of provider action to opportunistically screen individuals are contributing to the large number of hypertensives that are either unscreened or undiagnosed in rural Bihar. Interestingly, this study suggests that once individuals are aware of their high BP status, the vast majority access care. However, the quality of care received by individuals in rural Bihar is generally quite low across local provider types, which contributes to reduced rates of effective BP management and excess burden on patients who feel the need to travel great distances to receive quality care. Previous studies in similar contexts have demonstrated that hypertension can be diagnosed and treated at the community level [40]. The capacity of providers in rural Bihar, including IPs, to measure BP, diagnose hypertension, and stock and prescribe appropriate treatments must be improved within the existing legal constraints. Further, the role of community health workers, such as community health officers and accredited social health activists can be expanded in Bihar to bring services to populations. Future studies should seek to understand reasons for the perception of poor quality of care at public providers, even though PHCs often provide the highest quality of care locally available to patients. It is promising that coverage of hypertension management appears to be high in Bihar, now policymakers must urgently implement measures to improve the quality of these services in Bihar and similar contexts [41].

## Conclusion

Presenting supply-side, demand-side, and linked information about hypertension management is effective for demonstrating systemic gaps in disease control and helps to prioritize intervention design. This study's methodology should be considered for broader use in health systems research in other contexts and for other diseases. Findings suggest that while improving screening and diagnosis services would have the largest impact on retaining people in the care cascade, the quality of care provided in rural Bihar is low. With the rising burden of NCDs and hypertension in Bihar, increasing screening practices without first adequately equipping providers (of all types) would be futile to effectively manage the hypertension burden. Without first improving quality of care, patients newly linked to care are not likely to

experience health gains and may even suffer from adverse effects from the large proportion of providers who are unable to effectively provide hypertension management services.

## Supporting information

**S1 Annex. Assessment of primary health care in Bihar sample size calculations.** Provides motivations and justifications for parent study sample size.
(DOCX)

**S2 Annex. Provider quality cascade.** Provides definitions and data sources for steps along the provider quality cascade.
(DOCX)

**S3 Annex. Prescription rating protocol for clinical vignettes.** Provides the process undertaken by two clinicians for rating the appropriateness of prescriptions.
(DOCX)

**S4 Annex. Linking process for determining percent of hypertensive patients linked to quality care.** Provides a flow chart for the number of individuals and number of providers available at each level in the study, including the final number of hypertension patients linked to providers with known quality. The sample size of various regressions is also explained.
(DOCX)

**S5 Annex. Expanded hypertension cascade for adults age 30–54 in rural Bihar across two studies.** Provides a figure comparing the expanded hypertension care cascade between the Assessment of Primary Health Care in Bihar and the NFHS-4 studies.
(DOCX)

**S6 Annex. Inclusivity in global research.** Provide the completed inclusivity in global research questionnaire for this study.
(DOCX)

## Acknowledgments

We wish to acknowledge colleagues in the Government of Bihar, Oxford Policy Management, and CARE India; the enumerators and field team; and participants for their involvement in this research.

## Author Contributions

**Conceptualization:** Michael A. Peters, Olakunle Alonge, Anbrasi Edward, Krishna D. Rao.

**Data curation:** Japneet Kaur, Navneet Kumar.

**Formal analysis:** Michael A. Peters.

**Funding acquisition:** Krishna D. Rao.

**Methodology:** Michael A. Peters.

**Project administration:** Japneet Kaur.

**Resources:** Krishna D. Rao.

**Software:** Japneet Kaur.

**Supervision:** Michael A. Peters, Olakunle Alonge, Yvonne Commodore-Mensah, Krishna D. Rao.

**Validation:** Michael A. Peters, Anbrasi Edward, Yvonne Commodore-Mensah.

**Visualization:** Michael A. Peters.

**Writing – original draft:** Michael A. Peters.

**Writing – review & editing:** Michael A. Peters, Olakunle Alonge, Anbrasi Edward, Yvonne Commodore-Mensah, Japneet Kaur, Navneet Kumar, Krishna D. Rao.

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
