## [Decision Letter · Decision Letter 0]

4 Apr 2022

PGPH-D-22-00250

Barriers to Effective Hypertension Management in Rural Bihar, India: A cross-sectional, linked supply- and demand-side study

Dear Dr. Peters,

Thank you for submitting your manuscript to PLOS Global Public Health. After careful consideration, we feel that it has merit but does not fully meet PLOS Global Public Health’s publication criteria as it currently stands. Therefore, we invite you to submit a revised version of the manuscript that addresses the points raised during the review process.

We look forward to receiving your revised manuscript.

Kind regards,

Roopa Shivashankar, MD, MSc

Academic Editor

Journal Requirements:

1. Your co-authors:

Yvonne Commodore-Mensah -ycommod1@jhmi.edu

Japneet Kaur -japneetkaur58@gmail.com

Navneet Kumar -navneet.pcs@gmail.com

Krishna D Rao -kdrao@jhu.edu

,have not confirmed authorship of the manuscript. We have resent them the authorship confirmation email; however please check that the above email address for them is correct and follow up personally to ensure they confirm. 

Please note that we cannot proceed your manuscript  until we have received confirmations from all co-authors.

2. Please provide  separate figure files in .tif or .eps format only and remove any figures embedded in your manuscript file.  Please ensure that all files are under our size limit of 20MB.  

For more information about how to convert your figure files please see our guidelines: Once you've converted your files to .tif or .eps, please also make sure that your figures meet our format requirements:

3. Please amend your detailed Financial Disclosure statement. This is published with the article, therefore should be completed in full sentences and contain the exact wording you wish to be published.

ii). State the initials, alongside each funding source, of each author to receive each grant.

4. In the online submission form, you indicated that "The data used in this study are owned by a research consortium, led by the Government of Bihar. Permission has not been given by the government to deposit the raw data in a public repository. The research consortium (and government) has reviewed and approved the publication of the results included in this study.

The datasets used and analyzed in the current study are available for individual use upon approval from the research consortium. All requests may be directed to the corresponding author.". All PLOS journals now require all data underlying the findings described in their manuscript to be freely available to other researchers, either 1. In a public repository, 2. Within the manuscript itself, or 3. Uploaded as supplementary information.

Additional Editor Comments (if provided):

Reviewers' comments:

Reviewer's Responses to Questions

**Comments to the Author**

1. Does this manuscript meet PLOS Global Public Health’s publication criteria? Is the manuscript technically sound, and do the data support the conclusions? The manuscript must describe methodologically and ethically rigorous research with conclusions that are appropriately drawn based on the data presented.

Reviewer #1: Yes

Reviewer #2: Yes

2. Has the statistical analysis been performed appropriately and rigorously?

Reviewer #1: I don't know

Reviewer #2: Yes

3. Have the authors made all data underlying the findings in their manuscript fully available (please refer to the Data Availability Statement at the start of the manuscript PDF file)?

Reviewer #1: Yes

Reviewer #2: No

4. Is the manuscript presented in an intelligible fashion and written in standard English?

Reviewer #1: Yes

Reviewer #2: Yes

5. Review Comments to the Author

Reviewer #1: This research study describes the landscape of hypertension management in rural Bihar, India using linked supply- and demand-side estimates of quality care and receipt of care.

Detailed comments:

Line 130: people who seek care for chronic conditions, 39% receive care from private MBBS and AYUSH providers, and another 30% seek care from IPs. What about the remaining?

Line 219-221: “Measures of provider quality were described chronologically in terms of their impact on a patient’s progression through the hypertension care cascade, from screened to aware to treated and controlled BP.” As BP was not measured, not sure how the measure of BP control was evaluated.

Line 223: Definition of quality care needs to be revisited in the Indian context and provider type. The study investigators considered the criteria for “high quality HTN care” to include availability of first-line antiHTN medications. Majority of private doctors in India in single stand alone clinics do not stock antiHTN medications.

In addition, in India, AYUSH providers are not legally allowed to prescribe allopathic medications and hence may not be stocking drugs.

IP are essentially unqualified medical providers- again not legally allowed to stock nor prescribe medications

These factors need to be highlighted in the manuscript and may have impacted the assessment.

Line 296: Gives breakup of health seeking behaviour of the population. However, the proportion seeking care from IPs is not clear. Of the 94.4% population who sought treatment from the private sector, how many sought treatment with IPs as they were the largest provider population assessed?

Line 306: “Median monthly amount spent on HTN management was 250 rs.” Please give the breakup by provider as the OOPE would vary.

Line 361: "...people from the general caste were significantly more likely to receive quality care from a local provider than those from the SC caste." What was the health seeking behaviour by caste? Were the SCs more likely to seek care from IPs/ AYUsh- hence impacting the quality of care received (as measured under the study)?

Line 363: “Among individuals in the villages where quality of care was known, the number of quality providers in a village was not associated with the probability of bypassing local care options.” Not clear- please rephrase

Line 395: “…the quality-adjusted coverage of hypertension management services was as low as 10.9%.” This statement should be clarified in context of local providers.

Line 396: “Nearly 45% of individuals with reported hypertension bypassed local providers, traveling between 20 and 40 kilometers to receive care.” This statement is not aligned with Line 424 that states: “over a quarter of individuals (28.2%) traveled more than 20 kilometers to receive treatment”

Line 444: “The capacity of all providers in rural Bihar, including IPs, to measure BP, diagnose hypertension, and stock and prescribe appropriate treatments must be improved.” Need to underline policy options, legal barriers and various challenges esp with regard to AYUSH and IPs.

Line 462: Study concludes that patients may even suffer from adverse effects from the large proportion of providers who are unable to effectively provide hypertension management services. No data is presented on patients receiving harmful prescriptions that may cause adverse effects.

Other comments:

- IP- More background on informal providers (IPs) in the Indian context would be useful as they constituted more than 3/4ths of the total providers assessed

- Annexes 1-4 - not available

- Figure 1- not available

Reviewer #2: The authors have explored an interesting domain of NCD care (ie link between supply- and demand-side barriers) in rural Bihar. The manuscript has been prepared well. I have some minor queries and suggestions -

1. Please avoid the term 'allopathy' for what we consider medical sciences as either 'evidence-based healthcare/ medicine' or 'modern medicine'. I would prefer the former term. The oft-used term allopathy creates confusion among both scientific readers and laypeople, as it continues the notion of error-prone healthcare practices which were prevalent during the time the word 'allopathy' was popularized by proponents of 'homeopathy'. Moreover, as far as I remember, the National Medical Commission doesn't use the term 'allopathy' either in its curriculum or degree. Similarly, the names of degrees for different steam of AYUSH should be mentioned somewhere (eg BHMS for homeopathy).

2. The authors have missed introducing the NPCDCS well in their introduction. They have not considered the presence of Population-based screening (PBS) for selected NCDs and their risk factors which includes hypertension too. The standard treatment guidelines mentioned at reference number 33 is not appropriate for use in building your justification for screening all adults 18 years and above (line number 387-388). This is in contrast to what they themselves have done while including only adults 30 years and older only. This age group is being screened according to guidelines mentioned in both NPCDCS modules and HWC (Health and Wellness Centre Modules).

3. Though bypassing care provider is one of the key outcomes of the study, the same in missed in abstract's background/objective. Rather there is mention of supply- side barriers which I couldn't find sufficiently addressed in the introduction section or methods.

4. There seems to be mismatch between proportion of respondents mentioned to have received quality of care (31.0% in line number 40 vs 10.9% in line number 43).

5. Though the objectives in abstract mentions intent to examine supply- and demand-side barriers, authors have mentioned mostly about demand side barriers in introduction (line numbers 114-118).

6. Some of the references are inappropriate. For example, at line numbers 61-62, authors are arguing about hypertension being the leading CVD risk factor, but the references are unrelated mere narrative reviews or viewpoints and not scientific estimate studies or original articles. Similarly, I consider reference numbers 11, 16-18, 24, 32, 34 as either unrelated or inappropriate. In place of 24, identify a published reference from NSSO's most recent survey on health consumption pattern in India, since it has national-level estimates.

7. Authors have provided certain claims without any references. For example check line numbers 66-67, 76-77, 99-100, 163-164 (Demographic Health Service Provision Assessment Survey). Also, they should provide some reference for the ABPHC study/ report (line number 136).

8. I couldn't find reference number 12.

9. Please correct few incorrect reference styles (eg 6, 7, 12).

10. The rationale is related to the concept of 'healthcare provider switch behaviour'. The authors are encouraged to add a few references to this phenomenon in their introduction to justify their research idea. They also need to build upon the case of 'poor coverage' of screening for hypertension through PBS of NPCDCS as well as poor health system preparedness found in the most recent National NCD Monitoring Survey. This is important since the tools used in NNMS were adapted from internationally standardized WHO-SARA and WHO-PEN tools as compared to those used by the authors.

11. I have not seen 'study context' being part of methods section. Some of the sentences here should better be part of introduction. Authors should tell a bit of health care system of Bihar to the international audience.

12. I couldn't find the annexures, so I can't comment on them.

13. The unit of study need to be clarified in study design part (line number 142-143). Is it individuals or households? If it is individuals, then mentioning use of three-stage sampling design is wrong.

14. The reason for 30 as cluster size is not mentioned (line number 146). The basis for sample size calculation is not provided too.

15. How was the questionnaire developed? How was it validated?

16. The authors have mentioned about use of 'trained enumerators'. If multiple data collectors were use, it would be appreciated to mention about the inter-observer reliability during or after training to decrease risk of measurement bias.

17. Have the authors excluded specialist medical doctors (eg MD Medicine)? If yes, why?

18. I am not comfortable with the use of an unrelated facility readiness assessment tool for India. This is in contrast with what the NCD programme managers have been trained into.

19. A flow-chart for different numbers for study participants at different stage of study would be appropriate.

20. Give full description for Stata software (College Station Texas). Is there a license for this software with any of the study team authors? I

21. Check line number 320. Are AYUSH providers allowed to store medicines of modern medical sciences?

22. Check line numbers 348-349 and 359. In one place 74 people are mentioned to have their quality of care received known, while at other it is 73. Plz rectified this error.

23. Check line number 385-386. The national guideline is to screen for hypertension only once in a year. So why are you insisting on BP measurement at each visit to doctor?

24. Check line number 404-405, the claim that "local health systems are not providing adequate services to manage hypertension in Bihar" is ill-justified. The authors themselves have mentioned in limitation later that they couldn't contact the doctors outside the project villages who might have provided care to the study participants.

25. Check line numbers 396-97 and 424-25. There seem to be a typo or genuine error here?

26. Change the word 'biometric' with 'anthropometric' at line number 408.

27. Choice of visit to 'urban areas' to seek healthcare should not be considered in isolation for hypertension diagnosis or refill of medicines as the authors are arguing. The healthcare seeking decisions (especially for ambulatory care) in this part of India is often taken to piggy-back with other important household need for travel to town. Since, the authors have not explored reasons for 'bypassing care' and traveling out of village, the justification is at best an educated guess.

28. The role of CHOs (Community Health Officers) is not mentioned at all in hypertension screening, follow-up etc. This is surprising since the HWCs are being actively developed in almost all parts of India.

6. PLOS authors have the option to publish the peer review history of their article (what does this mean?). If published, this will include your full peer review and any attached files.

**Do you want your identity to be public for this peer review?** For information about this choice, including consent withdrawal, please see our Privacy Policy.

Reviewer #1: **Yes: **Anupam Khungar Pathni

Reviewer #2: **Yes: **Sanjeev Kumar

---

## [Decision Letter · Decision Letter 1]

4 Aug 2022

PGPH-D-22-00250R1

Barriers to Effective Hypertension Management in Rural Bihar, India: A cross-sectional, linked supply- and demand-side study

Dear Dr. Peters,

Thank you for submitting your manuscript to PLOS Global Public Health. After careful consideration, we feel that it has merit but does not fully meet PLOS Global Public Health’s publication criteria as it currently stands. Therefore, we invite you to submit a revised version of the manuscript that addresses the points raised during the review process.

We look forward to receiving your revised manuscript.

Kind regards,

Roopa Shivashankar, MD, MSc

Academic Editor

Journal Requirements:

Additional Editor Comments (if provided):

Reviewers' comments:

Reviewer's Responses to Questions

**Comments to the Author**

1. If the authors have adequately addressed your comments raised in a previous round of review and you feel that this manuscript is now acceptable for publication, you may indicate that here to bypass the “Comments to the Author” section, enter your conflict of interest statement in the “Confidential to Editor” section, and submit your "Accept" recommendation.

Reviewer #1: All comments have been addressed

Reviewer #2: All comments have been addressed

2. Does this manuscript meet PLOS Global Public Health’s publication criteria? Is the manuscript technically sound, and do the data support the conclusions? The manuscript must describe methodologically and ethically rigorous research with conclusions that are appropriately drawn based on the data presented.

Reviewer #1: Yes

Reviewer #2: Yes

3. Has the statistical analysis been performed appropriately and rigorously?

Reviewer #1: 

Reviewer #2: Yes

4. Have the authors made all data underlying the findings in their manuscript fully available (please refer to the Data Availability Statement at the start of the manuscript PDF file)?

Reviewer #1: No

Reviewer #2: Yes

5. Is the manuscript presented in an intelligible fashion and written in standard English?

Reviewer #1: Yes

Reviewer #2: Yes

6. Review Comments to the Author

Reviewer #1: The authors have addressed most of the reviewers' comments.

A few minor comments:

1. Line 174: “The public sector infrastructure includes about 1,800 Primary Health Centers (PHC), of which 455 are fully operational.” Please provide reference for the statement

2. Line 280: Please clarify what is implied by “initiated treatment upon diagnosis”. Does this refer to only medication as many patients with grade 1 hypertension may be started on lifestyle measures at diagnosis.

3. Line 456: Hypertension prevalence is mentioned as 22%. Citation is incorrect to the data presented. May consider citing Geldsetzer et al.

4. Line 473: “most individuals perceived the quality of care at PHCs to be poor”. Did the study assess the availability of staff at the PHCs that could have contributed to this perception?

5. Line 481: Blood pressure measurement is not an anthropometric measure. Maybe the authors want to state “anthropometric measurements and BP”?

Reviewer #2: I am satisfied with the responses by the authors.

7. PLOS authors have the option to publish the peer review history of their article (what does this mean?). If published, this will include your full peer review and any attached files.

**Do you want your identity to be public for this peer review?** For information about this choice, including consent withdrawal, please see our Privacy Policy.

Reviewer #1: No

Reviewer #2: **Yes: **Sanjeev Kumar

---

## [Editor Report · Decision Letter 2]

23 Aug 2022

PGPH-D-22-00250R2

Barriers to Effective Hypertension Management in Rural Bihar, India: A cross-sectional, linked supply- and demand-side study

Dear Dr. Peters,

Thank you for submitting your manuscript to PLOS Global Public Health. After careful consideration, we feel that it has merit but does not fully meet PLOS Global Public Health’s publication criteria as it currently stands. Therefore, we invite you to submit a revised version of the manuscript that addresses the points raised during the review process.

We look forward to receiving your revised manuscript.

Kind regards,

Roopa Shivashankar, MD, MSc

Academic Editor

Journal Requirements:

Additional Editor Comments (if provided):

Please provide details in the legend of table-4. It is unclear what is presented. For example the odd ratio of the age groups are thousands which are unlikely. Provide clarification for the same. 
---

## [Editor Report · Decision Letter 3]

16 Sep 2022

Barriers to Effective Hypertension Management in Rural Bihar, India: A cross-sectional, linked supply- and demand-side study

PGPH-D-22-00250R3

Dear Dr Peters,

We are pleased to inform you that your manuscript 'Barriers to Effective Hypertension Management in Rural Bihar, India: A cross-sectional, linked supply- and demand-side study' has been provisionally accepted for publication in PLOS Global Public Health.

Best regards,

Roopa Shivashankar, MD, MSc

Academic Editor